# Effect of Oral Health Education Using a Mobile App (OHEMA) on the Oral Health and Swallowing-Related Quality of Life in Community-Based Integrated Care of the Elderly: A Randomized Clinical Trial

**DOI:** 10.3390/ijerph182111679

**Published:** 2021-11-07

**Authors:** Ji-Yun Ki, Se-Rim Jo, Kyung-Sook Cho, Jung-Eun Park, Ja-Won Cho, Jong-Hwa Jang

**Affiliations:** 1Department of Public Health, Graduate School, Dankook University, Cheonan-si 31116, Korea; dhkjy2296@gmail.com (J.-Y.K.); serim0222@naver.com (S.-R.J.); cucukxx@hanmail.net (K.-S.C.); 2Department of Dental Hygiene, College of Health Science, Dankook University, Cheonan-si 31116, Korea; jepark@dankook.ac.kr; 3Department of Preventive Dentistry, College of Dentistry, Dankook University, Cheonan-si 31116, Korea; priscus@dankook.ac.kr

**Keywords:** elderly, oral care, oral education, home care, xerostomia, swallowing-related quality of life

## Abstract

This study investigated the effect of oral health education using a mobile app (OHEMA) on the oral health and swallowing-related quality of life (SWAL-QoL) of the elderly population in a community-based integrated care project (CICP). Forty elderly individuals in the CICP were randomized into intervention and control groups. OHEMA provided information on customized oral health care management, oral exercises, and intraoral and extraoral massage methods for 50 min/session, once a week, for 6 weeks. Pre- and post-intervention surveys assessed the unstimulated salivary flow rate, subjective oral dryness, tongue pressure, and SWAL-QoL, which were analyzed using ANCOVA and repeated measures ANOVA. In the intervention group, tongue pressure increased significantly from pre- (17.75) to post-intervention (27.24) (*p* < 0.001), and subjective oral dryness decreased from pre- (30.75) to post-intervention (18.50). The unstimulated salivary flow rate had a higher mean score in the intervention group (7.19) than in the control group (5.04) (*p* < 0.001). The SWAL-QoL significantly improved from pre- (152.10) to post-intervention (171.50) in the intervention group (*p* < 0.001) but did not change significantly in the control group (*p* > 0.05). OHEMA appears to be a useful tool for oral health education for the elderly as it improved the SWAL-QoL, with increased tongue pressure and reduced oral dryness.

## 1. Introduction

South Korea is predicted to become a super-aged society by 2025 [1], and the national health care cost burden is rapidly increasing, particularly in relation to health issues associated with the elderly population. This has highlighted the care of the elderly as a political issue, and a national-level community-based integrated care project (CICP) has been in operation since 2019 [2]. The CICP is a social service system that provides public health care and services that meet the needs of individual residents of the community who require care, to enable them to achieve personal desires and perform activities as cooperating members of the community [2]. The scope of the CICP includes anyone that requires care in South Korea, but the largest proportion of care is directed to elderly people who have been discharged from a hospital, most of whom are reported to have a systemic disease and a low level of oral health [3].

The reported criteria of oral frailty include oral dryness (xerostomia); motor dysfunction of the tongue, mouth, and lips; low tongue pressure; and reduced swallowing function [4]. Notably, among oral functions, the decrease in tongue pressure could lead to swallowing dysfunction [5]. More than half of the community-resident elderly are in the high-risk dysphagia group, with an increased risk of problems related to food ingestion, such as anorexia, which can cause nutritional imbalances and have a negative impact on systemic health [6]. Oral health care management is essential for enhancing basic health and quality of life in senescence [7,8]. It has been reported that oral dryness, mouth muscle function, and swallowing are positively influenced by continuous oral health care and education [9].

The South Korean government has gradually extended the CICP as a national public health policy [2]. As a result, the demand for the visiting oral care program of the CICP has increased amongst the elderly [10]. However, currently, there are only a few oral health care programs available, while a certain number of existing programs are available to the elderly at care hospitals [11,12,13,14,15]. A recent study showed that oral massage and oral exercises performed on the elderly in care hospitals increased the salivary flow rate, thereby reducing oral dryness and enhancing the swallowing-related quality of life (SWAL-QoL) [13,14]. In addition, oral exercise with gum was reported to improve oral function, including chewing ability, saliva flow rate, and swallowing [16], while mouth gymnastics was shown to improve the swallowing function in the elderly [17].

Most oral health educational material used in earlier studies included posters, PowerPoint presentations, and demonstrations [13,14,18]. The recent coronavirus disease 2019 (COVID-19) pandemic has led to the extended application of video education materials because of the establishment of a non-face-to-face culture [19]. Notably, mobile-based video education programs were found to have positive effects in improving participants’ knowledge, education-related satisfaction, and practice of the educational content [20,21,22]. The reports also stated that approximately 50% of the educational content was received by the participants, compared with that from the other educational materials [23], and an especially strong effect on the education of individual elderly persons was observed as the programs could be watched repeatedly without time or space constraints [19,24]. The oral health education using a mobile app (OHEMA) program for the CICP visiting care of the elderly is developed in this study, and future interventions that apply the OHEMA program are anticipated to improve oral health self-care and the practice of oral functional training activities, thus contributing to enhanced oral health [3]

Thus, the purpose of this study was to investigate the effects of the OHEMA program on the unstimulated salivary flow rate (USFR), subjective oral dryness, tongue pressure, and SWAL-QoL in the elderly in the CICP by measuring their health and oral health status.

## 2. Materials and Methods

### 2.1. Study Design and Participants

This study was conducted as a randomized clinical trial. The number of elderly people aged ≥ 65 years, living in Cheonan-si, Chungcheongnam-do, South Korea, was approximately 73,517, of whom 767 were enrolled in the CICP program. Of the 767 participants, 100 were included for visiting oral health care. The selection criteria were patients with systemic disease or the elderly who sought professional oral health care whether they experienced oral discomfort or not. Among them, 46 participants in the OHEMA program were selected [25]. The inclusion criteria were understanding of the content and purpose of the study; voluntary agreement to participate; ability to communicate without linguistic, auditory, or visual impairment; and normal cognitive ability. The exclusion criteria were missing two or more sessions of the program; a rate of practice of oral health behaviors less than 80%; and a history of systemic disease that could influence oral health, such as administration of a drug affecting saliva secretion, Sjögren syndrome, oral cancer, and stroke. The selected subjects were randomized between the intervention and control groups. The sample size was estimated as 38 using the G*Power program, based on the following conditions: two measurements through repeated measures ANOVA, effect size of 0.25, significance level of 0.05, and testing power of 0.85 [26]. Considering a 20% drop-out rate, 46 participants (intervention group n = 24, control group n = 22) were recruited. The final analysis involved 40 individuals (intervention group n = 20, control group n = 20) who participated in both the pre- and post-intervention surveys (Figure 1).

This study complied with the guidelines of the Helsinki Declaration for the ethical treatment of participants and was conducted with the approval of the Institutional Review Board of the D University (IRB: DKU 2020-05-014-001). Informed consent was obtained from all the participants involved in this study.

The study data were registered in the Clinical Research Information Service (CRIS) as part of the World Health Organization (WHO) International Clinical Trials Registry Platform (trial registration number: KCT0005990).

### 2.2. Intervention 

#### 2.2.1. OHEMA

In the OHEMA intervention, a video playing a trot song (3 min) for oral health education was provided in the introduction to motivate the CICP elderly participants regarding the importance of oral health. The main educational content was shown through four videos consisting of oral exercise with gum and a clock ticking sound (7 min), intraoral and extraoral massage (6 min), and a customized oral hygiene intervention including brushing and denture care methods (15 min). In addition, a workbook and a poster with identical content were provided to the homes [27] (Figure 2). Oral health education media used for OHEMA are presented in the Appendix A).

#### 2.2.2. Intervention Group 

The intervention group, after participating in the pre-intervention survey, received the OHEMA intervention for 6 weeks and then participated in the post-intervention survey. The authors helped participants download and trained them on the use of the mobile app before they received OHEMA. The OHEMA program was provided by a member of the research group who visited the home of the participant for customized 1:1 education (50 min per session) based on the participant’s characteristics. The intervention group was also guided to practice the oral health education content described in the poster on their own every day and record it on the self-checklist, while the visiting research member checked the level of practice at each visit.

#### 2.2.3. Control Group

The control group did not undergo any oral hygiene education or OHEMA intervention. However, the same pre-/post-intervention tests were performed as in the intervention group, and oral health education similar to that in the intervention group was provided after the posttest.

### 2.3. Variable Measures

The OHEMA intervention was performed between 1 February and 30 April 2021, and the data were collected through survey interviews and direct oral examinations. The survey questionnaire was based on a previous study [3] and consisted of general characteristics of the participants (sex, age, smoking, drinking, education, living status, and systemic disease) and oral health behavior (oral examination, oral health education, brushing patterns for a day (yesterday), number of brushings per day, oral care products, and denture use). Oral frailty was assessed based on subjective oral dryness, perceived chewing disability, chewing disability, swallowing difficulty, and inconvenient pronunciation [28,29]; dietary factors (meal types, mealtimes, and daily amount of water); activities of daily living (ADL) [30,31,32] and instrumental activities of daily living (IADL) [33]; mini-mental state examination—Korean version (MMSE-K) [34]; the multidimensional scale of perceived social support (MSPSS) [35,36]; and oral health-related quality of life (OHRQoL) scales (oral health impact profile (OHIP-14), geriatric oral health assessment index (GOHAI), and SWAL-QoL) [37,38,39,40,41,42]. For oral health status, the number of remaining functional teeth, plaque index (PI) [43], tongue coating index [44], USFR [13], and tongue pressure [45] were measured. For intra- and inter-rater reliability, the authors conducted training before the beginning of the study and confirmed a concordance of > 90%. The results for the intraclass correlation coefficient (ICC) to test the reliability of the dependent variables were as follows: subjective oral dryness (ICC = 0.82; 95% CI = 0.65–0.87), tongue pressure (ICC = 0.85; 95% CI = 0.72–0.92), USFR (ICC = 0.89; 95% CI = 0.80–0.95), and SWAL-QoL (ICC = 0.75; 95% CI = 0.53–0.87), indicating a high level of reliability.

#### 2.3.1. Clinical Oral Examination Assessment

The assessment of oral health status used the WHO [46] criteria, and the number of remaining functional teeth was determined during an oral examination. 

The PI was assessed by applying a disclosing solution (Dharma Research, Inc., Miami, FL, USA) to every tooth and evaluating the colored plaque on each tooth surface with a maximum score of 5 and a score of 0 for tooth surfaces without color, with higher scores indicating lower levels of oral health self-care [43]. The tongue coating index was assessed with a score of 1 for the presence and 0 for the absence of tongue coating, with the tongue divided into nine sections of the same width from the tip of the tongue; higher total scores indicated lower levels of oral health care [44]. The USFR was assessed by positioning a Schirmer strip (WF41-1850) cut to the size of 1 × 17 cm on the dorsal surface of the tongue and measuring the level of wetness of the strip after 1 min of lightly closing the mouth [13]. Tongue pressure was assessed as the maximum pressure on the JMS tongue pressure device (JMS Co., Ltd., Hiroshima, Japan) in relation to the mean tongue pressure of 30.0 kPa for individuals aged ≥ 65 years, with a tongue pressure of 20 kPa or below indicating a significantly high risk of diaphragmatic spasm or aspiration pneumonia [45].

#### 2.3.2. Physical and Cognitive Ability Assessment 

To measure the level of daily activity of the participants, the ADL [30,31,32] with six categories and the IADL [33] with eight categories were used. The scores ranged between 1 and 3 based on the level of performance, with higher scores indicating lower levels of independent daily activity. In this study, the reliability of ADL was Cronbach’s *a* = 0.76 and that of IADL was Cronbach’s *a* = 0.87.

To measure the level of cognitive ability, the MMSE-K was used [34]. The MMSE-K comprises six domains and nine categories to suit the elderly in South Korea, assessing orientation to time and place, memory registration, attention and calculation, memory recall, linguistic function, comprehension, and determination. The maximum score is 30, and higher scores indicate stronger cognitive abilities. Reliability in this analysis was Cronbach’s *a* = 0.77. 

The MSPSS measures family support, friend support, and special support on a 5-point Likert scale (1 = strongly disagree/5 = strongly agree), with a maximum score of 60 and higher scores indicating higher levels of social support [35,36]. In this analysis, reliability was Cronbach’s *a* = 0.74.

#### 2.3.3. Subjective Oral Health Assessment

The categories of perceived chewing disability, chewing disability, swallowing difficulty, and inconvenient pronunciation, which are used to indicate the level of oral frailty in the National Oral Health Nutrition Investigation, were measured on a 5-point Likert scale (1 = strongly disagree/5 = strongly agree), where mean values were obtained, and higher scores indicated higher levels of oral frailty [28]. For subjective oral dryness, the scale of six categories developed by Lee et al. [29] based on a visual analogue scale (1 = strongly disagree/10 = strongly agree) was used, where the maximum score was 60 and higher scores indicated higher levels of oral dryness. In this study, reliability was Cronbach’s *a* = 0.80.

#### 2.3.4. Quality of Life Assessment

The OHIP-14 [37] consists of 14 categories based on a 5-point Likert scale (1 = strongly disagree/5 = strongly agree) to measure the OHRQoL; the maximum score is 70 and higher scores indicate a higher OHRQoL. Reliability in this analysis was Cronbach’s *a* = 0.91.

The GOHAI consists of 12 categories based on a 5-point Likert scale (1 = strongly disagree/5 = strongly agree) to measure the quality of life related to oral health; the maximum score is 60 and higher scores indicate a higher OHRQoL [38,39,40]. Reliability in this analysis was Cronbach’s *a* = 0.81.

SWAL-QoL consists of 44 categories based on a 5-point Likert scale (1 = strongly disagree/5 = strongly agree) to measure the quality of life related to swallowing; the maximum score is 220 and higher scores indicate a higher QoL [41,42]. Reliability in this analysis was Cronbach’s *a* = 0.95.

### 2.4. Statistical Analysis

Data were analyzed using the SPSS program (IBM SPSS Statistics 23.0 for Windows, SPSS Inc., Chicago, IL, USA). The normality of the analyzed variables was tested using the Shapiro–Wilk test. The homogeneity of the intervention and control groups was tested using the independent *t*-test, the chi-square test, and Fisher’s exact test. To compare the main variables pre- and post-intervention, the paired *t*-test was used, while the effect size was analyzed using ANCOVA and two-way repeated measure ANOVA. The significance level was set to α = 0.05.

## 3. Results

### 3.1. Homogeneity Test 

#### 3.1.1. Homogeneity of the General Characteristics, and Physical and Cognitive Functions

Table 1 presents the results of the homogeneity assessment of the general characteristics of participants. Between the intervention and control groups, homogeneity was verified with respect to sex, age, smoking and drinking, education, living status, systemic disease, and ADL, IADL, MMSE-K, and MSPSS scores (*p* > 0.05). Notably, an important consideration for OHEMA for the elderly is the level of daily activity or reduction in cognitive ability. The ADL and IADL scores of the participants showed a mild level of disability in both the intervention (1.71 and 12.14, respectively) and control (0 and 10.67, respectively) groups, but the level did not pose a constraint to participation in oral health education or oral health care performance. In addition, both the intervention and control groups showed normal scores (≥24) in the MMSE-K in cognitive functional assessment with known cut-off and diagnostic validity.

#### 3.1.2. Homogeneity of Oral Health and Dietary Factors

PI, swallowing difficulty, and mealtime in the intervention group were significantly higher than in the control group (Table 2). Regarding the oral health behavior between the two groups, homogeneity was verified for oral examination, oral health education, brushing for a day (yesterday), number of brushings per day, and oral care products (*p* > 0.05). In terms of oral health status, between-group homogeneity was verified for the number of functional teeth, tongue coating index, perceived chewing disability, chewing disability, and inconvenient pronunciation, as well as OHIP-14 and GOHAI (*p* > 0.05). Among the dietary factors, homogeneity was verified for the meal type and amount of water per day (*p* > 0.05).

#### 3.1.3. Homogeneity of Dependent Variables

Table 3 presents the results of the homogeneity of the dependent variables of participants. Between-group homogeneity was verified for subjective oral dryness, tongue pressure, and SWAL-QoL (*p* > 0.05), but the USFR was lower in the intervention group (4.02) than in the control group (5.56) (*p* = 0.003).

### 3.2. Effects of OHEMA on Oral Health and SWAL-QoL

The results of comparisons of the dependent variables before and after the 6-week OHEMA program are presented in Table 4.

For subjective oral dryness, the scores significantly decreased from pre- (30.75) to post-intervention (18.50) in the intervention group (*p* < 0.001), with a significant between-group interaction in time verifying the effect of OHEMA in reducing subjective oral dryness (*p* < 0.001). For tongue pressure, the scores significantly increased from pre- (17.75) to post-intervention (27.24) (*p* < 0.001) in the intervention group, with a significant between-group interaction in time (*p* < 0.001).

The USFR displayed significant variation in the homogeneity test. Analyzing the USFR as a covariate prior to intervention showed the significant effect of the intervention when the measurements before and after intervention were compared (*p* < 0.001). The estimated values controlling the preliminary measurements were higher in the intervention group (7.19) than in the control group (5.04) with statistical significance (*p* < 0.001), thus verifying the positive effect of OHEMA. In addition, analysis of the SWAL-QoL as a covariate prior to intervention, based on the between-group heterogeneity in swallowing disability, showed a significant increase in SWAL-QoL from pre- (152.10) to post-intervention (171.50) in the intervention group (*p* < 0.001), with a significant between-group interaction in time, thus verifying the effect of OHEMA in enhancing the SWAL-QoL (*p* = 0.012).

## 4. Discussion

With accelerated population aging, the demand for visiting oral care management has increased, as public health care policies, through the CICP, receive increased attention to overcome the limitations in the care of the elderly in care hospitals [47,48,49]. The COVID-19 pandemic has also led to more robust non-face-to-face video education programs, highlighting the need for video-based oral health education of the elderly that effectively provides educational content and does not have time or space constraints [19,23,24]. Thus, this study analyzed the impact of OHEMA application for 6 weeks on the oral health and SWAL-QoL of the selected CICP elderly population, which was randomized between intervention and control groups.

Most variables, including general characteristics, physical and cognitive functioning, and oral health and dietary factors, showed verified homogeneity between the intervention and control groups. The mean age of the participants in this study was 79 years, and although the ADL and IADL did not vary between the two groups, the scores were lower than those in a previous study of the elderly [48], presumably because of the distribution of elderly individuals aged ≥ 75 years between the intervention (65%) and control (70%) groups and the fact that in all groups, 90% of the elderly had a systemic disease. Nevertheless, all participants showed a level of daily activity that did not prevent participation in the program. In particular, the MMSE-K scores were within the normal range, and the level of education was higher than that of elementary school graduation in 90% and 80% of the participants in the intervention and control groups, respectively, indicating a suitable level of cognitive ability and education that would allow a positive effect from oral health education [47,48]. 

Among the oral health behaviors, the level of unmet dental checkups was high in the intervention (65%) and control (75.0%) groups, which agreed with the 67.2% of unmet dental care in the CICP visiting care elderly reported previously [3]. The proportion of denture use was 45.0% in the intervention group and 55.0% in the control group, which agreed with the percentage of denture users (50%) in the study by Jang et al. [3]. The mean number of functional teeth was 19.5 in the intervention group and 14.9 in the control group. The mean number of functional teeth in the participants among the CICP visiting care elderly was 14.92, a level higher than that reported by Jang et al. [3], presumably because the level of oral health was slightly higher in the participants that satisfied the inclusion criteria in this study in comparison to the general participants.

As an indicator of oral frailty, the perceived chewing disability was slightly high in both the intervention (3.40) and control (3.75) groups, while the tongue pressure in the preliminary oral examination was very low in both the intervention (17.75) and control (17.78) groups. This coincides with the high risk of diaphragmatic spasm or aspiration pneumonia at a tongue pressure of 20 kPa in the elderly aged ≥ 65 years [45]. However, the increase in tongue pressure to 27.24 kPa in the intervention group after the 6-week OHEMA was proof of its positive effect, with a predicted effect of enhancing the SWAL-QoL. According to a related earlier study, the muscular strength of the anterior tongue improved after oral exercise intervention [50], supporting the results of this study and verifying the positive effect of oral exercise on tongue pressure in the elderly. A decrease in tongue pressure is one of the oral health problems in the elderly and is deeply correlated with reduced swallowing function and nutritional imbalance, with a potential impact on the physical health of the whole body [42,51]. Thus, oral exercise education is essential in maintaining the basic physical health of the elderly.

Meanwhile, the subjective oral dryness in the intervention group after the 6-week OHEMA decreased from pre- (30.75) to post-intervention (18.50), while the USFR increased from 4.02 to 7.19. These results coincided with the results of earlier studies on oral health care intervention in the elderly [13,52,53]. In addition, Thanga Raj et al. [54] reported that oral exercise could help reduce oral dryness, based on which OHEMA is predicted to increase the SFR to reduce oral dryness and potentially affect food ingestion in the elderly.

Previous studies on quality of life and health behaviors in the community-resident elderly reported a high correlation with all health behaviors of an individual to protect, maintain, and improve his or her health [55,56]. In this study, the OHIP-14 and GOHAI scores were at a moderate level pre-intervention, and the SWAL-QoL significantly increased from pre- (152.10) to post-intervention (171.50). However, the scores were lower than those from an earlier study on the quality of life in the healthy elderly (170.73) [57], which is presumed to be due to the markedly lower level of oral health in the CICP elderly, most of whom had a history of hospital admission to treat a systemic disease [3]. The reduced swallowing function is caused by the decrease in chewing ability and weakening of the mouth muscles [16,17]. The swallowing function is deeply correlated with social activities and lifestyle, such as dietary habits [41], with a significant impact on the overall quality of life [57]. It is also closely related to mortality [58]. Providing continuous health care interventions, such as OHEMA, to improve the swallowing function is likely to enhance the quality of life in senescence. 

In conclusion, the effectiveness of OHEMA in the visiting oral care CICP elderly has been verified in this study, based on which the program may be nationally implemented. In addition, as the first study of a CICP-based intervention in the field of oral health care, this study lends support to the effective use of programs, such as OHEMA, in future visiting oral care projects. In contrast to the conventional elderly oral health education, which is mainly composed of demonstrations [13,14,18], the use of audio–visual materials such as videos and posters, which promote self-learning, and the consequent established self-learning in this study were shown to be effective in reducing oral dryness and enhancing the SWAL-QoL. In addition, the mobile app videos used in the study intervention were made available on YouTube so that the participants could repeat the learning without time or place constraints [20,22,24]. OHEMA is thus anticipated to allow self-motivated learning through repeated watching of the videos even when direct visits of the educators are not possible, while notable significance lies in its potential for expanded application to all elderly participants, including those in the CICP.

Nevertheless, there were several limitations in this study. First, the participants were the CICP elderly among the residents of certain communities, so the potential selection bias prevents generalization to all elderly people in the nation. Second, despite the significant results of enhanced oral health and SWAL-QoL in the intervention group after OHEMA, the program was applied for only 6 weeks, and the long-term effects of the program cannot be readily verified. Third, the SWAL-QoL was measured using a questionnaire with subjective responses, and actual, objective measurements were not taken. Fourth, the oral health knowledge, attitude, and behavior adjustment before and after the OHEMA could not be measured.

Therefore, further studies should categorize the elderly participants based on the characteristics of each individual group and develop a customized program accordingly. It is also necessary to develop a program that contains a variety of mobile app-based content for different circumstances the participants might encounter using a multidisciplinary approach. In addition, the study design should include a longitudinal study with long-term monitoring, as well as objective, quantitative measurements of the various psychosocial factors and oral health improvement effects. It is essential to conduct multidisciplinary research and development of suitable educational content and methods to improve oral health care knowledge, attitudes, and behavior in the elderly of different residential types. In addition, we suggest that the OHEMA program used in this study be applied to a larger number of participants to further identify changes in oral health behaviors.

## 5. Conclusions

The 6-week application of OHEMA in the CICP elderly was shown to increase tongue pressure, reduce oral dryness, and improve the SWAL-QoL. Thus, OHEMA is predicted to be useful in activities to promote oral health in the elderly of the CICP or at care hospitals. This study was the first to verify the effects of OHEMA in improving the oral health of the CICP elderly, and the results collectively suggest that the program could be used in visiting oral care services for the elderly in accordance with the national public health care policy, with an extended scope for nationwide implementation from 2026. Nevertheless, follow-up studies and extended studies for verifying the duration of the effects and counteracting the limitations should be conducted. 

## Figures and Tables

**Figure 1 ijerph-18-11679-f001:**
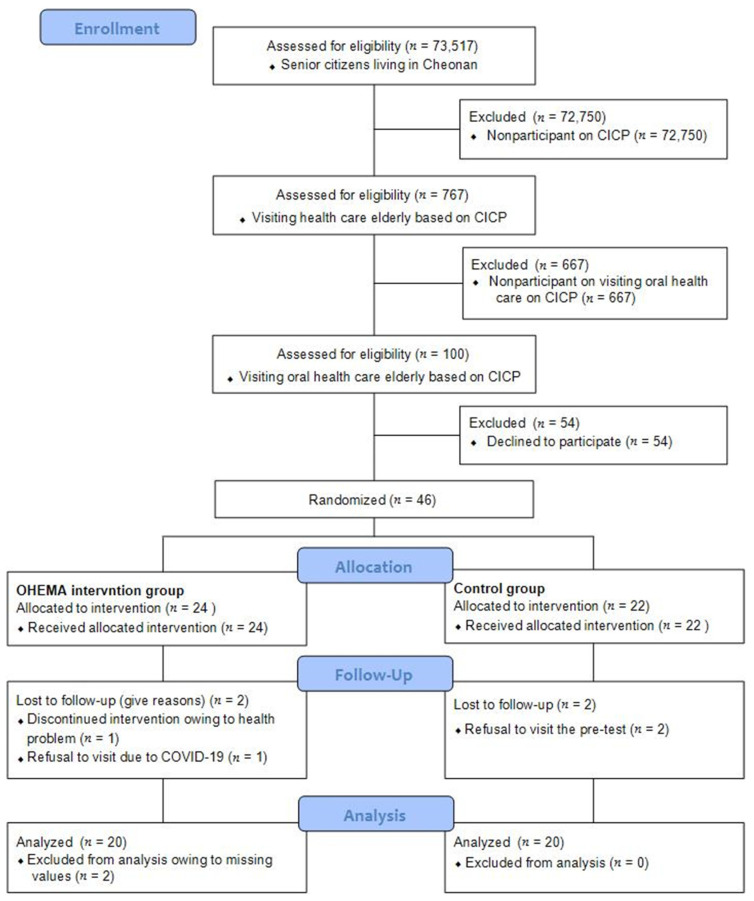
Consolidated standards of reporting trials (CONSORT) diagram of participants.

**Figure 2 ijerph-18-11679-f002:**
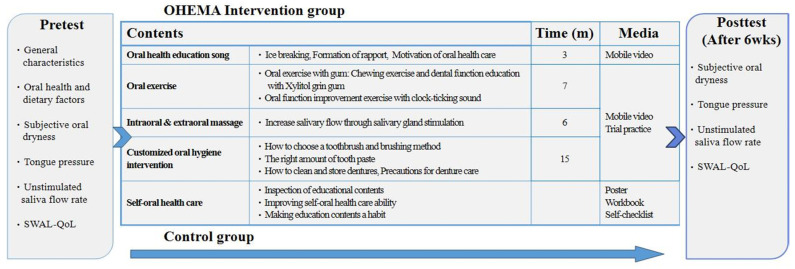
Flowchart of oral health education intervention using mobile app (OHEMA).

**Table 1 ijerph-18-11679-t001:** Homogeneity of general characteristics of participants.

Characteristics	Categories	Intervention (*n* = 20)	Control (*n* = 20)	*p*-Value *
Sex	Male	10 (50.0)	7 (35.0)	0.523
	Female	10 (50.0)	13 (65.0)	
Age	65–74	7 (35.0)	6 (30.0)	0.080
	≥75	13 (65.0)	14 (70.0)	
Smoking	Yes	2 (10.0)	3 (15.0)	1.000
	No	18 (90.0)	17 (85.0)	
Drinking	Yes	3 (15.0)	4 (20.0)	1.000
	No	17 (85.0)	16 (80.0)	
Education (years)	None	2 (10.0)	4 (20.0)	0.588
	6	11 (55.0)	8 (40.0)	
	≥7	7 (35.0)	8 (40.0)	
Living alone	Yes	11 (55.0)	10 (50.0)	0.285
	No	9 (45.0)	10 (50.0)	
Systemic disease	No	2 (10.0)	2 (10.0)	1.000
	Yes	18 (90.0)	18 (90.0)	
ADL		1.47 ± 2.37	1.94 ± 3.02	0.614
IADL		11.79 ± 4.24	14.00 ± 4.32	0.137
MMSE-K		24.92 ± 1.83	24.08 ± 2.31	0.339
MSPSS		44.70 ± 9.29	43.05 ± 7.13	0.532

Data are presented as *n* (%) or mean ± standard deviation; * chi-square test and Fisher’s exact test or independent *t*-test at α = 0.05; ADL = activities of daily living; IADL = instrumental activities of daily living; MMSE-K = mini-mental state examination—Korean version; MSPSS = multidimensional scale of perceived social support.

**Table 2 ijerph-18-11679-t002:** Homogeneity of oral health and dietary factors between the intervention and control groups.

Factors	Characteristics	Categories	Intervention (*n* = 20)	Control (*n* = 20)	*p*-Value *
Oral health behavior	Oral examination	Yes	7 (35.0)	5 (25.0)	0.731
		No	13 (65.0)	15 (75.0)	
	Oral health education	Yes	14 (70.0)	13 (65.0)	1.000
		No	6 (30.0)	7 (35.0)	
	Brushing for a day yesterday	Yes	20 (100.0)	19 (95.0)	1.000
		No	0 (0)	1 (5.0)	
	Number of brushing/d	<3	7 (35.0)	14 (70.0)	0.056
		≥3	13 (65.0)	6 (30.0)	
	Oral care products	Use	3 (15.0)	6 (30.0)	0.451
		Unuse	17 (85.0)	14 (70.0)	
	Denture use	Yes	9 (45.0)	11 (55.0)	0.752
		No	11 (55.0)	9 (45.0)	
Oral health status	Number of functional teeth		19.50 ± 10.14	14.90 ± 10.87	0.174
	Plaque index		2.51 ± 0.86	1.78 ± 1.18	0.049
	Tongue coating index		3.70 ± 2.15	3.35 ± 2.52	0.639
Oral frailty	Perceived chewing disability		3.40 ± 1.27	3.75 ± 1.16	0.370
	Chewing disability		2.60 ± 1.31	2.40 ± 1.19	0.616
	Difficulty of swallowing		3.20 ± 1.06	2.10 ± 1.21	0.004
	Inconvenient pronunciation		3.00 ± 1.17	2.95 ± 1.23	0.896
Oral health-related quality of life	OHIP-14		48.75 ± 12.80	52.80 ± 12.00	0.308
	GOHAI		37.15 ± 10.31	39.20 ± 8.92	0.505
Dietary factors	Meal type	PR, PS	16 (80.0)	16 (80.0)	1.000
		PR, MS	2 (10.0)	2 (10.0)	
		P, P/MS	2 (10.0)	2 (10.0)	
	Mealtime (min)		22.00 ± 8.34	16.55 ± 8.02	0.042
	Amount of water (cup)/d		5.75 ± 2.73	5.55 ± 3.46	0.840

Data are presented as n (%) or mean ± standard deviation; * chi-square test and Fisher’s exact test or independent *t*-test at α = 0.05; OHIP-14 = oral health impact profile; GOHAI = geriatric oral health assessment index; PR = plain rice; PS = plain side dish; MS = minced side dish; P = porridge.

**Table 3 ijerph-18-11679-t003:** Homogeneity of dependent variables between the intervention and control groups.

Characteristics	Intervention (*n* = 20)	Control (*n* = 20)	t	*p*-Value *
Subjective oral dryness	30.75 ± 14.97	27.15 ± 13.16	0.809	0.424
Tongue pressure (kPa)	17.75 ± 8.46	17.78 ± 7.15	−0.014	0.989
USFR (mm)	4.02 ± 1.55	5.56 ± 1.48	−3.222	0.003
SWAL-QoL	152.10 ± 6.83	169.30 ± 6.92	1.837	0.074

Data are presented as mean ± standard deviation; * independent *t*-test at α = 0.05; USFR = unstimulated salivary flow rate; SWAL-QoL = swallowing-related quality of life scale.

**Table 4 ijerph-18-11679-t004:** Comparison of oral health and swallowing-related quality of life between the intervention and control groups.

Characteristics	Pretest	Posttest	*p*-Value *	Between-Group *p*-Value **
Subjective oral dryness				
Intervention (*n* = 20)	30.75 ± 3.15	18.50 ± 2.82	<0.001	<0.001
Control (*n* = 20)	27.15 ± 3.15	29.25 ± 2.82	0.176	
Tongue pressure (kPa)				
Intervention (*n* = 20)	17.75 ± 8.46	27.24 ± 6.62	<0.001	<0.001
Control (*n* = 20)	17.78 ± 1.75	17.90 ± 1.65	0.858	
USFR (mm)				
Intervention (*n* = 20)	4.02 ± 1.55	7.19 ± 0.19	<0.001	<0.001
Control (*n* = 20)	5.56 ± 1.48	5.04 ± 0.19	0.409	
SWAL-QoL				
Intervention (*n* = 20)	152.10 ± 6.83	171.50 ± 6.82	0.002	<0.001
Control (*n* = 20)	169.85 ± 6.83	163.30 ± 6.92	0.265	

Data are presented as mean ± standard error; * paired *t*-test; ** ANCOVA test for USFR and SWAL-QoL and two-way repeated measure ANOVA for all other variables at α = 0.05; USFR = unstimulated salivary flow rate; SWAL-QoL = swallowing-related quality of life scale.

## Data Availability

The data presented in this study are available on reasonable request from the corresponding author.

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
