# Peer review of "Effect of Oral Health Education Using a Mobile App (OHEMA) on the Oral Health and Swallowing-Related Quality of Life in Community-Based Integrated Care of the Elderly: A Randomized Clinical Trial"

_ijerph, 2021, doi:10.3390/ijerph182111679_

Round 1

Reviewer 1 Report

This manuscript reports effects of oral health education using a mobile app in older adults.

As the population ages, maintaining and improving the health of the older adults is an important in any situation. Therefore, this survey has an important role.

Major comments
1)  The word 「elderly」 is used in your manuscript, but please check if there is any difference in meaning.

2) 2.2 Intervention

The author says the purpose of this study was to investigate the effect of OHEMA education program. To compare the effectiveness of the educational programs, the authors need to explain a little more detail about the control group and the apps. Did the control group receive oral hygiene education only once before the study? The OHEMA intervention group believes the results will vary depending on whether you need to access the APP yourself or if you have a display that encourages you to use the app. For notification-prompting devices, the results may be based on whether or not practice has been done, rather than comparing the effects of an educational program.

Author Response

Reviewer 1

This manuscript reports effects of oral health education using a mobile app in older adults.

As the population ages, maintaining and improving the health of the older adults is an important in any situation. Therefore, this survey has an important role.

Thank you for reviewing our research. Below are our responses to your comments and queries. We have made every effort to incorporate your recommendations into the revised manuscript. Our revised paper has been checked by a native English speaker (American Journal Experts).

Major comments

1) The word 「elderly」 is used in your manuscript, but please check if there is any difference in meaning.

Authors’ response:

We appreciate your valuable suggestion. Participants in this study were referred to as ‘Elderly’ because they were subjects of Community-based integrated care and were over the age of 65. In Korea, based on the ‘Elderly Welfare Act’, the age of 65 or older is defined as “elderly”.

2) 2.2 Intervention

The author says the purpose of this study was to investigate the effect of OHEMA education program. To compare the effectiveness of the educational programs, the authors need to explain a little more detail about the control group and the apps. Did the control group receive oral hygiene education only once before the study? The OHEMA intervention group believes the results will vary depending on whether you need to access the APP yourself or if you have a display that encourages you to use the app. For notification-prompting devices, the results may be based on whether or not practice has been done, rather than comparing the effects of an educational program.

Authors’ response:

Thank you very much for your detailed assessment and kind advice. In response to your valuable comments, we have added a description of the control group and the app.

  • 2.2.2. Intervention group

The authors helped participants download and trained them on the use of the mobile app before they received OHEMA.

  • 2.2.3. Control group

The control group did not undergo any oral hygiene education or OHEMA intervention. However, the same pre-posttests were performed as in the intervention group, and oral health education similar to that in the intervention group was provided after the post-test.

We have made our best efforts to accommodate your recommendations in the revised manuscript. Please let us know in detail if you have any further recommendations for modifications. We would be glad to incorporate any further revisions required. Thank you very much.

Reviewer 2 Report

First of all congratulations for this interesting topic, it was a pleasure to review your manuscript. I think that such projects will be very benefitial, as more and more people, who are over 65 are using mobile applications. There is a big potential to create such program, especially during these COVID-19 times.

The manuscript is well organised, and the language editing service was great. All the structure is professional, just like the data evaluation. 

My only concern is the sample size. I think you might include much more people, especially as this is a mobile app trial. 100 people is extremely low number, I would suggest you next time this number should be minimum 500, because otherwise I feel this is just scientifically "raping the data". 

Maybe for the next manuscript! I suggest to publish it!

Author Response

Reviewer 2

First of all congratulations for this interesting topic, it was a pleasure to review your manuscript. I think that such projects will be very benefitial, as more and more people, who are over 65 are using mobile applications. There is a big potential to create such program, especially during these COVID-19 times.

The manuscript is well organised, and the language editing service was great. All the structure is professional, just like the data evaluation.

My only concern is the sample size. I think you might include much more people, especially as this is a mobile app trial. 100 people is extremely low number, I would suggest you next time this number should be minimum 500, because otherwise I feel this is just scientifically "raping the data".

Maybe for the next manuscript! I suggest to publish it!

Authors’ response:

Thank you very much for your constructive comment. This study was an intervention study to evaluate the effectiveness of oral health promotion using the oral health education mobile app in visiting oral care, based on the community integrated care project. The study was conducted by analyzing the appropriate sample size in this study design, and significant changes in oral health were confirmed.

In the next study, we plan to attempt to verify the effect of oral health education using a mobile app on oral health behavior by expanding the number of participants on a larger scale.

We have added the following to the Discussion section to reflect your valuable opinions.

  • In addition, we suggest that the OHEMA program used in this study be applied to a larger number of participants to further identify changes in oral health behaviors.

We have made our best efforts to accommodate your recommendations in the revised manuscript. Please let us know in detail if you have any further recommendations for modifications. We would be glad to incorporate any further revisions required. Thank you very much. 

Reviewer 3 Report

With the COVID-19 pandemic, social activities are restricted, so oral function is expected to decline. Therefore, I think it is very important to develop an educational program to maintain and improve oral function. In this point, this study is very interesting and useful. However, some points should be revised.

1. Line 74: Please describe the full name of OHEMA.

2. Lines 85-86: "Of the 767 participants, 100 were included in the OHEMA program." Are there any criteria for participating in the OHEMA program? Do you encourage participants with impaired oral function to participate in this program? Or is it due to the voluntary intention of the participants?

3. Line 133: Please describe the intra- and inter-reliability of each variable measures.

4. Table 2: Plaque index, difficulty of swallowing, and meal time in the  intervention group were significantly higher than in the control group. Please describe these results in the "Results 3.1.2" section.

Author Response

Reviewer 3

With the COVID-19 pandemic, social activities are restricted, so oral function is expected to decline. Therefore, I think it is very important to develop an educational program to maintain and improve oral function. In this point, this study is very interesting and useful. However, some points should be revised.

Thank you for reviewing our research. Below are our responses to your comments and queries. We have made every effort to incorporate your recommendations into the revised manuscript. Our revised paper has been checked by a native English speaker (American Journal Experts).

1. Line 74: Please describe the full name of OHEMA.

Authors’ response:

We appreciate your valuable suggestion. We have revised as follow:

  • “oral health education using a mobile app (OHEMA)”

2. Lines 85-86: "Of the 767 participants, 100 were included in the OHEMA program." Are there any criteria for participating in the OHEMA program? Do you encourage participants with impaired oral function to participate in this program? Or is it due to the voluntary intention of the participants?

Authors’ response:

We appreciate the Reviewer’s valuable suggestion and we agree. As shown in Figure 1, the Community Integrity Care Project (CICP) is a leading national project, with 767 people participating in various health and welfare programs. We have revised the sentence "Of the 767 participants, 100 were included in the OHEMA program." as follows:

  • “Of the 767 participants, 100 were included for visiting oral health care. The selection criteria were patients with systemic disease or the elderly who sought professional oral health care, even if they experienced oral discomfort or not. Among them, 46 participants in the OHEMA program were selected.”

As already mentioned in the text, 46 participants in the OHEMA program were randomly assigned to an intervention group (n = 24) or a control group (n = 22) and the selection and exclusion criteria were previously described in the text.

3. Line 133: Please describe the intra- and inter-reliability of each variable measures.

Authors’ response:

Thank you for your constructive comment. We have added Cronbach’s s a for some variables,  and the intra-and inter-reliability of each of the variables measured in our study as follows.:

  • Lines 153-158: ‘For intra- and inter-reliability, the authors conducted training before the beginning of the study and confirmed the concordance of > 90%. The results for the intraclass correlation coefficient (ICC) to test the reliability of the dependent variables were as follows: subjective oral dryness (ICCs = 0.82; 95% CI = 0.65-0.87), tongue pressure (ICCs = 0.85; 95% CI = 0.72-0.92), USFR (ICCs = 0.89; 95% CI = 0.80-0.95), and SWAL-QoL (ICCs = 0.75; 95% CI = 0.53-0.87), indicating a high level of reliability.’
  • Lines 179-180: In this study, the reliability of ADL was Cronbach’s a = 0.76 and that of IADL was Cronbach’s a = 0.87.
  • Lines 185-186: Reliability in this analysis was Cronbach’s a = 0.77.

4. Table 2: Plaque index, difficulty of swallowing, and meal time in the intervention group were significantly higher than in the control group. Please describe these results in the "Results 3.1.2" section.

Authors’ response:

Thank you for your valuable comment. In response to your comment, we have revised the content as follows:

  • ‘PI, swallowing difficulty, and mealtime in the intervention group were significantly higher than in the control group (Table 2).’

We have made our best efforts to accommodate your recommendations in the revised manuscript. Please let us know in detail if you have any further recommendations for modifications. We would be glad to incorporate any further revisions required. Thank you very much.

Round 2

Reviewer 1 Report

The manuscript has been revised well.

I think this manuscript will be acceptable.